# Diverse Client Selection for Federated Learning via Submodular Maximization

**Ravikumar Balakrishnan**[*]
Intel Labs
ravikumar.balakrishnan@intel.com

**Tian Li**[*]
CMU
tianli@cmu.edu

**Tianyi Zhou**[*]
University of Washington
tianyizh@uw.edu

**Nageen Himayat**
Intel Labs
nageen.himayat@intel.com

**Virginia Smith**
CMU
smithv@cmu.edu

**Jeffrey Bilmes**
University of Washington
bilmes@uw.edu

## Abstract

In every communication round of federated learning, a random subset of clients communicate their model updates back to the server which then aggregates them all. The optimal size of this subset is not known and several studies have shown that typically random selection does not perform very well in terms of convergence, learning efficiency and fairness. We, in this paper, propose to select a small diverse subset of clients, namely those carrying representative gradient information, and we transmit only these updates to the server. Our aim is for updating via only a subset to approximate updating via aggregating all client information. We achieve this by choosing a subset that maximizes a submodular facility location function defined over gradient space. We introduce "federated averaging with diverse client selection (DivFL)". We provide a thorough analysis of its convergence in the heterogeneous setting and apply it both to synthetic and to real datasets. Empirical results show several benefits of our approach, including improved learning efficiency, faster convergence, and more uniform (i.e., fair) performance across clients. We further show a communication-efficient version of DivFL that can still outperform baselines on the above metrics.

## 1 Introduction

Federated learning (FL) involves collaboratively training of machine learning model across a large number of clients while keeping client data local. Recent approaches to this problem repeatedly alternate between device-local (stochastic) gradient descent steps and server-aggregation of the clients' model updates (McMahan et al., 2017). In cross-device settings, a server and its model usually serve several thousands of devices. Therefore, the communication between clients and the server can be costly and slow, forming a huge impediment to FL's viability.

One property of the collection of clients that can mitigate these problems, however, is often not exploited, and that is redundancy. Specifically, many clients might provide similar, and thus redundant, gradient information for updating the server model. Therefore, transmitting all such updates to the server is a waste of communication and computational resources. How best to select a representative and more informative client set while adhering to practical constraints in federated learning is still an open challenge. Although several selection criteria have been investigated in recent literature, e.g., sampling clients with probabilities proportional to their local dataset size (McMahan et al., 2017), sampling clients of larger update norm with higher probability (Chen et al., 2020), and selecting clients with higher losses (Balakrishnan et al., 2020; Cho et al., 2020), the redundancy and similarity of the clients' updates sent to the server is not represented and exploited in these approaches. In particular, communicating multiple clients' updates to the server may cause statistical and system inefficiency if too many of them are too similar to each other. The commonly studied modular score/probability for each individual client is incapable of capturing information as a property over a *group* of clients.

---

[*]Equal contributions

Ideally, a diverse set of clients would be selected, thereby increasing the impact of under-represented clients that contribute different information, and thereby improving fairness. This, in fact, is a topic of increasing interest (Mohri et al., 2019; Cho et al., 2020; Dennis et al., 2021; Huang et al., 2021).

In this paper, we introduce diversity to client selection in FL, namely a strategy to measure how a selected subset of clients can represent the whole when being aggregated on the server. Specifically, in each communication round, we aim to find a subset whose aggregated model update approximates the aggregate update over all clients. By doing this, we aim to limit the impact of subset selection which introduces variance in the model updates across round, that could otherwise slow the learning process. Inspired by the CRAIG method of coreset selection for efficient machine learning training (Mirza-soleiman et al., 2020), we derive an upper bound of the approximation error as a supermodular set function (in particular, the min-form of the facility location function (Cornuéjols et al., 1977)) evaluated on the selected subset. We can then apply submodular maximization (Fujishige, 2005; Iyer et al., 2013; Wei et al., 2014) on a complement submodular function to (approximately) minimize the error upper bound. We employ the greedy selection (Nemhauser et al., 1978) of a subset of clients according to the marginal gain of the submodular function to achieve a solution with provable approximation guarantee (Conforti & Cornuejols, 1984). By integrating the diverse client selection into the most commonly studied FL scheme, i.e., Federated Averaging (FedAvg) (McMahan et al., 2017), we propose `DivFL` that applies global model aggregation over a selected subset of clients after multiple local steps on every client. We present theoretical convergence analysis of `DivFL` and show its tolerance to the heterogeneity of data distributions across clients and large numbers of local steps. However, our method differs from the CRAIG method where selection is performed based on model updates (involving multiple epochs at the clients). In addition, our approach allows for partial device participation where the server does not have access to all data at any communication round, as is standard in FL (McMahan et al., 2017). In experiments, we compare `DivFL` with other client selection approaches on both synthetic dataset and FEMNIST, wherein our method excels on convergence, fairness, and learning efficiency.

## 2 BACKGROUND AND RELATED WORK

We consider a typical federated learning objective:

$$\min_{w} f(w) = \sum_{k=1}^{N} p_k F_k(w),$$

where for each client $k \in [N]$, $p_k$ is a pre-defined weight (such that $\sum_{k=1}^{N} p_k = 1$) that can be set to $\frac{1}{N}$ or the fraction of training samples, and $F_k$ is the client-specific empirical loss. While there are various possible modeling approaches, we consider this canonical objective of fitting a single global model to the non-identically distributed data across all clients (McMahan et al., 2017).

**Client Selection in Federated Learning.** Client[1] sampling is a critical problem particularly for cross-device settings where it is prohibitive to communicate with all devices. Two common (or default) strategies are (a) sampling the clients based on the number of local data points and uniformly averaging the model updates, and (b) sampling the clients uniformly at random and aggregating the model updates with weights proportional to the local samples (Li et al., 2020). There is also recent work proposing advanced sampling techniques to incorporate dynamic systems constraints, accelerate the convergence of federated optimization, or to obtain a better model with higher accuracy (Nishio & Yonetani, 2019; Ribero & Vikalo, 2020; Cho et al., 2020; Lai et al., 2020). We investigate client selection through the lens of encouraging client diversity at each communication round which largely remains unexplored in previous work. The closest client selection method to ours is based on clustering (e.g., selecting representative clients from separate clusters (Dennis et al., 2021)). We note that performing (private) clustering in federated settings is still an open problem, and our method can be viewed as a soft version of dynamic clustering at each round (discussed in the next paragraph). The benefits of gradient (or model) diversity has been demonstrated in other related contexts, such as scaling up mini-batch stochastic gradient descent (SGD) (Yin et al., 2018). Enforcing sample or gradient diversity during optimization also implicitly places more emphasis on the underrepresented

---

[1]Following conventions, we use the term 'client' for the problem of client selection. Throughout the paper, we use 'devices' and 'clients' interchangeably.

sub-population of clients, and can promote fairness defined as representative disparity (Hashimoto et al., 2018). Similar to previous work (e.g., Cho et al., 2020; Balakrishnan et al., 2020), we observe our approach yields more fair solutions across the network in Section 5.

**Diverse Subset Selection via Submodularity.** Modular scores have been widely studied for subset selection in machine learning and federated learning, e.g., a utility score for each sample or client often measured by the loss. However, the diversity of a subset cannot be fully captured by such modular scores since there is no score interaction. Diversity is often well modeled by a diminishing return property, i.e., the (marginal) gain an element brings to a subset diminishes as more elements added to the subset. There exists a rich and expressive family of functions, all of which are natural for measuring diversity, and all having the diminishing returns property: given a finite ground set $V$ of size $n$, and any subset $A \subseteq B \subseteq V$ and a $v \notin B$, a set function $F : 2^V \to \mathbb{R}$ is submodular if

$$F(v \cup A) - F(A) \geq F(v \cup B) - F(B). \tag{1}$$

This implies $v$ is no less valuable to the smaller set $A$ than to the larger set $B$. The marginal gain of $v$ conditioned on $A$ is denoted $f(v|A) \triangleq f(v \cup A) - f(A)$ and reflects the importance of $v$ to $A$. Submodular functions (Fujishige, 2005) have been widely used for diversity models (Lin & Bilmes, 2011; Batra et al., 2012; Prasad et al., 2014; Gillenwater et al., 2012; Bilmes & Bai, 2017).

Maximizing a submodular function usually encourages the diversity and reduces the redundancy of a subset. This property has been utilized for data selection in active learning (Guillory & Bilmes, 2011), curriculum learning (Zhou & Bilmes, 2018), mini-batch partitioning (Wang et al., 2019), gradient approximation (Mirzasoleiman et al., 2020), etc. Although the number of possible subsets $A$ is $\binom{n}{k}$, enumerating them all to find the maximum is intractable. Thanks to submodularity, fast approximate algorithms (Nemhauser et al., 1978; Minoux, 1978; Mirzasoleiman et al., 2015) exist to find an approximately optimal $A$ with provable bounds (Nemhauser et al., 1978; Conforti & Cornuejols, 1984). Despite its success in data selection, submodularity has not been explored for client selection in federated learning. Encouraging diversity amongst local gradients (or model updates) of selected clients can effectively reduce redundant communication and promote fairness. Moreover, it raises several new challenges in the FL setting, e.g., (1) it is unclear which submodular function to optimize and in which space to measure the similarity/diversity between clients; (2) What convergence guarantee can be obtained under practical assumptions such as heterogeneity among clients, and (3) What are the effects of outdated client selection due to communication constraints?

## 3 DIVERSE CLIENT SELECTION

In this section, we introduce "federated averaging with diverse client selection" (or `DivFL`), a method that incorporates diverse client selection into the most widely studied FL scheme, federated averaging (FedAvg). We will first derive a combinatorial objective for client selection via an approximation of the full communication from all clients, which naturally morphs into a facility location function in the gradient space that can be optimized by submodular maximization. We then present the standard greedy algorithm that optimizes the objective by selecting a diverse subset of clients at every communication round.

### 3.1 APPROXIMATION OF FULL COMMUNICATION

We aim to find a subset $S$ of clients whose aggregated gradient can approximate the full aggregation over all the $N$ clients $V = [N]$. To formulate this problem, we start by following the logic in Mirzasoleiman et al. (2020). Given a subset $S$, we define a mapping $\sigma : V \to S$ such that the gradient information $\nabla F_k(v^k)$ from client $k$ is approximated by the gradient information from a selected client $\sigma(k) \in S$. For $i \in S$, let $C_i \triangleq \{k \in V | \sigma(k) = i\}$ be the set of clients approximated by client-$i$ and $\gamma_i \triangleq |C_i|$. The full aggregated gradient can be written as

$$\sum_{k \in [N]} \nabla F_k(v^k) = \sum_{k \in [N]} \left[ \nabla F_k(v^k) - \nabla F_{\sigma(k)}(v^{\sigma(k)}) \right] + \sum_{k \in S} \gamma_k \nabla F_k(v^k). \tag{2}$$

Subtracting the second term from both sides, taking the norms, and applying triangular inequality, we can obtain an upper bound for the approximation to the aggregated gradient by $S$, i.e.,

$$\left\| \sum_{k \in [N]} \nabla F_k(v^k) - \sum_{k \in S} \gamma_k \nabla F_k(v^k) \right\| \leq \sum_{k \in [N]} \left\| \nabla F_k(v^k) - \nabla F_{\sigma(k)}(v^{\sigma(k)}) \right\|. \tag{3}$$

The above inequality holds for any feasible mapping $\sigma$ since the left hand side does not depend on $\sigma$. So we can take the minimum of the right-hand side w.r.t. $\sigma(k)$, $\forall k \in [N]$, i.e.,

$$\left\| \sum_{k \in [N]} \nabla F_k(v^k) - \sum_{k \in S} \gamma_k \nabla F_k(v^k) \right\| \leq \sum_{k \in [N]} \min_{i \in S} \left\| \nabla F_k(v^k) - \nabla F_i(v^i) \right\| \triangleq G(S). \quad (4)$$

The right hand side provides a relaxed objective $G(S)$ for minimizing the approximation error on the left hand. Minimizing $G(S)$ (or maximizing $\bar{G}$, a constant minus its negation) equals maximizing a well-known submodular function, i.e., the facility location function (Cornuéjols et al., 1977). To restrict the communication cost, we usually limit the number of selected clients to be no greater than $K$, i.e., $|S| \leq K$. This resorts to a submodular maximization problem under cardinality constraint, which is NP-hard but an approximation solution with $1 - e^{-1}$ bound can be achieved via the greedy algorithm (Nemhauser et al., 1978).

## 3.2 Greedy Selection of Clients

The naïve greedy algorithm for minimizing the upper bound of gradient approximation starts from $S \leftarrow \emptyset$, and adds one client $k \in V \backslash S$ with the greatest marginal gain to $S$ in every step, i.e.,

$$S \leftarrow S \cup k^*, \ \ k^* \in \underset{k \in V \backslash S}{\operatorname{argmax}} [\bar{G}(S) - \bar{G}(\{k\} \cup S)] \quad (5)$$

until $|S| = K$. Although it requires evaluating the marginal gain for all clients $k \in V \backslash S$ in every step, there exists several practical accelerated algorithms (Minoux, 1978; Mirzasoleiman et al., 2015) to substantially reduce the number of clients participating in the evaluation. For example, stochastic greedy algorithm (Mirzasoleiman et al., 2015) selects a client $k^*$ from a small random subset of $V \backslash S$ with size $s$ in each step, i.e.,

$$S \leftarrow S \cup k^*, \ \ k^* \in \underset{k \in \operatorname{rand}(V \backslash S, \ \operatorname{size}=s)}{\operatorname{argmax}} [\bar{G}(S) - \bar{G}(\{k\} \cup S)] \quad (6)$$

To incorporate the client selection into any federated learning algorithm, we apply the stochastic greedy algorithm in each aggregation round and perform selection only from a random subset of *active* clients in the network. The complete procedure is given in Algorithm 1 assuming the base algorithm is Federated Averaging (FedAvg) (McMahan et al., 2017).

On the left-hand side of Eq. (3)-(4), we aim at approximating the full communication by a weighted sum over selected clients in $S$ with weights $\{\gamma_i\}_{i \in S}$. However, since we relax the problem to minimizing its upper bound and the (stochastic) greedy solution is not guaranteed to achieve the global minimum of the relaxed objective, the weight associated with the greedy solution $S$, i.e., $\gamma_i = |C_i|$ with $C_i = \{k \in V | i \in \arg\min_{j \in S} \left\| \nabla F_k(v^k) - \nabla F_j(v^j) \right\|\}$, is sub-optimal. In fact, given $S$, the optimal weight $\{\gamma_i\}_{i \in S}$ can be achieved by directly minimizing the left hand side of Eq. (3)-(4) but it is infeasible because the full aggregation $\sum_{k \in [N]} \nabla F_k(v^k)$ is not available in our setting. Though there might exist better choices, we find that simple uniform weights work promisingly in all evaluated scenarios of our experiments. The stochastic greedy selection in line 2 of Algorithm 1

---

**Algorithm 1** `DivFL`

**Input:** $T, E, \eta, w_0$

1 **for** $t = 0, \cdots, T - 1$ **do**
2    Server selects a subset of $K$ active clients $S_t$ using the stochastic greedy algorithm in Eq. (6), and sends $w_t$ to them.
3    **for** *device $k \in S_t$ in parallel* **do**
4       $w^k \leftarrow w_t$
5       Solve the local sub-problem of client-$k$ inexactly by updating $w^k$ for $E$ local mini-batch SGD steps:
$$w^k = w^k - \eta \nabla F_k(w^k)$$
6       Send $\Delta_t^k := w_t^k - w_t$ back to Server
7    **end**
8    Server aggregates $\{\Delta_t^k\}$:
$$w_{t+1} \leftarrow w_t + \frac{1}{|S_t|} \sum_{k \in S_t} \Delta_t^k$$
9 **end**
10 **return** $w_T$

---

requires access to the gradients from all clients, which might be expensive in communication costs. In practice, we may take several approaches to minimize the communication costs. One is to receive periodic gradient updates from all clients every $m$ communication rounds. Another is to only use the

gradients from the selected clients at the current round to update part of the $N \times N$ dissimilarity matrix. In our evaluation, we take the latter method with "no-overheads". While this maybe suboptimal since a large part of the dissimilarity matrix will contain stale gradients, we observe no significant loss in performance in our empirical studies.

## 4 CONVERGENCE ANALYSIS

In this section, we provide theoretical analyses of the convergence behavior of Algorithm 1 for strongly convex problems under practical assumptions of non-identically distributed data, partial device participation, and local updating. While the current analysis only holds for the proposed client selection algorithm applied to FedAvg, it can be naturally extended to other federated optimization methods as well.

As discussed in Section 3.1, we draw connections between full gradient approximation and submodular function maximization. By solving a submodular maximization problem in the client selection procedure, we effectively guarantee that the approximation error is small (see Eq. (4)). We state an assumption on this below.

**Assumption 1** (Gradient approximation error). *At each communication round $t$, we assume the server selects a set $S_t$ of devices such that their aggregated gradients (with weights $\{\gamma_k\}_{k \in S_t}$) is a good approximation of the full gradients on all devices with error $\epsilon$, i.e.,*

$$\left\| \frac{1}{N} \sum_{k \in S_t} \gamma_k \nabla F_k(v_t^k) - \frac{1}{N} \sum_{k \in [N]} \nabla F_k(v_t^k) \right\| \leq \epsilon.$$

The same assumption has been studied in previous works on coreset selection for mini-batch SGD (Mirzasoleiman et al., 2020, Theorem 1). Note that $\epsilon$ is used as a measure to characterize how good the approximation is and our theorem holds for any $\epsilon < \infty$. Our algorithm (Algorithm 1) is effectively minimizing an upper bound of $\epsilon$ to achieve a potentially small value via running submodular maximization to select diverse clients (Eq. (4)). Next, we state other assumptions used in our proof, which are standard in the federated optimization literature (e.g., Li et al., 2019).

**Assumption 2.** *Each $F_k$ ($k \in [N]$) is $L$-smooth.*

**Assumption 3.** *Each $F_k$ ($k \in [N]$) is $\mu$-strongly convex.*

**Assumption 4.** *For $k \in [N]$ and all $t$, in-device variance of stochastic gradients on random samples $\zeta$ are bounded, i.e., $\mathbb{E}[\|\nabla F_k(w_t^k, \zeta) - \nabla F_k(w_t^k)\|^2] \leq \sigma^2$.*

**Assumption 5.** *For $k \in [N]$ and all $t$, the stochastic gradients on random samples $\zeta$ are uniformly bounded, i.e., $\|\nabla F_k(w_t^k, \zeta)\|^2 \leq G^2$.*

**Assumption 6** (Bounded heterogeneity). *Statistical heterogeneity defined as $F^* - \sum_{k \in [N]} p_k F_k^*$ is bounded by $C$, where $F^* := \min_w f(w)$ and $F_k^* := \min_v F_k(v)$.*

Let $w^* \in \arg\min_w f(w)$ and $v_k^* \in \arg\min_v F_k(v)$ for $k \in [N]$. In our analysis, we need to bound another variant of heterogeneity $\|\sum_{k \in [N]} p_k v_k^* - w^*\|$ in the parameter space. Note that under Assumption 3 ($\mu$-strongly convexity), Assumption 6 implies that $\|\sum_{k \in [N]} p_k v_k^* - w^*\|$ is also bounded by a constant.

**Setup.** Following Li et al. (2019), we flatten local SGD iterations at each communication round, and index gradient evaluation steps with $t$ (slightly abusing notation). We define *virtual* sequences $\{v_t^k\}_{k \in [N]}$ and $\{w_t^k\}_{k \in [N]}$ where

$$v_{t+1}^k = w_t^k - \eta_t \nabla F_k(w_t^k), \quad w_{t+1}^k = \begin{cases} v_{t+1}^k, & \text{if not aggregate,} \\ \text{select } S_{t+1} \text{ and average } \{v_{t+1}^k\}_{k \in S_{t+1}}, & \text{otherwise.} \end{cases}$$

While all devices virtually participate in the updates of $\{v_t^k\}$ at each *virtual* iteration $t$, the effective updating rule of $\{w_t^k\}$ is the same as that in Algorithm 1. Further, let $\overline{v}_t := \sum_{k \in [N]} p_k v_t^k, \overline{w}_t := \sum_{k \in [N]} p_k w_t^k$. Therefore, $\overline{w}_t = \begin{cases} \overline{v}_t & \text{if not aggregate,} \\ \frac{1}{K} \sum_{k \in S_t} v_t^k & \text{otherwise.} \end{cases}$. The goal of defining $\overline{v}_t$ and $\overline{w}_t$

is to relate the updates of $\overline{v}_t$ to mini-batch SGD-style updates, and relate the actual updates of $\overline{w}_t$ to those of $\overline{v}_t$. We aim at approximating $\overline{v}_{t+1}$ by $\overline{w}_{t+1}$ (when aggregating) and next state a main lemma bounding $\|\overline{w}_{t+1} - \overline{v}_{t+1}\|$.

**Lemma 1.** *For any virtual iteration $t$, under Algorithm 1 and Assumptions 1-6, we have*

$$\|\overline{w}_{t+1} - \overline{v}_{t+1}\| \leq LGE(E-1)\eta_{t_0}^2 + E\epsilon\eta_{t_0},$$

*where $L$ is the smoothness parameter, $G$ is the bounded stochastic gradient parameter, $E$ is the number of local iterations, and $\eta_{t_0}$ is indexing the latest communication round.*

We defer the proof to Appendix A. The main step involves using Assumption 1 to bound the approximate error of gradients and relating accumulative gradients with model updates. With Lemma 1, we state our convergence results as follows.

**Theorem 1** (Convergence of Algorithm 1). *Under Assumptions 1-6, we have*

$$\mathbb{E}[\|w^* - w_t\|^2] \leq O(1/t) + O(\epsilon).$$

The non-vanishing term $\epsilon$ encodes the gradient approximation error (Assumption 1), and will become zero when we select all clients at each round (i.e., $K = N$). In experiments, we observe that DivFL allows us to achieve faster convergence (empirically) at the cost of additional solution bias (a non-diminishing term dependent on $\epsilon$).

We provide a sketch of the proof here and defer complete analysis to Appendix A. Examine the distances between $\overline{w}_{t+1}$ and $w^*$,

$$\|\overline{w}_{t+1} - w^*\|^2 = \|\overline{w}_{t+1} - \overline{v}_{t+1}\|^2 + \|\overline{v}_{t+1} - w^*\|^2 + 2\langle\overline{w}_{t+1} - \overline{v}_{t+1}, \overline{v}_{t+1} - w^*\rangle.$$

If iteration $t$ is not an aggregation step, $\overline{w}_{t+1} = \overline{v}_{t+1}$ and

$$\|\overline{w}_{t+1} - w^*\|^2 = \|\overline{v}_{t+1} - w^*\|^2,$$

which we can bound with Lemma 1 in Li et al. (2019):

$$\mathbb{E}[\|\overline{v}_{t+1} - w^*\|^2] \leq (1 - \eta_t\mu)\mathbb{E}[\|\overline{w}_t - w^*\|^2] + \eta_t^2 B \tag{7}$$

for some constant $B$. If $t$ is an aggregation step, we need to bound

$$\mathbb{E}[\|\overline{w}_{t+1} - \overline{v}_{t+1}\|^2] + \mathbb{E}[\|\overline{v}_{t+1} - w^*\|^2] + 2\mathbb{E}[\langle\overline{w}_{t+1} - \overline{v}_{t+1}, \overline{v}_{t+1} - w^*\rangle].$$

The second term can be bounded by Eq. (7), which contains $(1 - \eta_t\mu)\mathbb{E}[\|\overline{w}_t - w^*\|^2]$. Therefore, combined with Lemma 1, with a decaying step size, we can obtain a recursion on $\mathbb{E}[\|\overline{w}_{t+1} - w^*\|^2]$ which leads to Theorem 1. We provide the complete proof in Appendix A.

## 5 EXPERIMENTS

**Setup.** We evaluate the DivFL approach utilizing both synthetic and real federated datasets from the LEAF federated learning benchmark (Caldas et al., 2019). The synthetic dataset enables us to control the data heterogeneity across clients for evaluation. We consider two baselines to compare our DivFL against: a) random sampling without replacement, and b) the power-of-choice approach (Cho et al., 2020) where clients with the largest training losses are selected. For DivFL, we consider an ideal setting where 1-step gradients are queried from every device in the network for each global round. In addition, we also evaluate the "no overhead" setting where (a) client updates from previous rounds are utilized to update part of the dissimilarity matrix, and (b) we run the stochastic greedy algorithm to avoid selecting from an entire set of clients. While the former provides the upper bound on the performance, the "no overhead" approach and other variants are more suited to realistic settings. For all the methods, we fix the number of clients per round $K = 10$ and report the performance for other choices of $K$ in Appendix. Each selected client performs $\tau = 1$ round of local model update before sharing the updates with the server unless otherwise noted. We report the performance of DivFL across metrics including convergence, fairness and learning efficiency. We further report the impact of the subset size $K$ on these metrics. Our code is publicly available at github.com/melodi-lab/divfl.

## 5.1 RESULTS ON THE SYNTHETIC DATASET

We generate synthetic data following the setup described in Li et al. (2020). The parameters and data are generated from Gaussian distributions and the model is logistic regression. $y = \arg\max(\text{softmax}(\mathbf{W}^T\mathbf{X} + \mathbf{b}))$. We consider a total of 30 clients where the local dataset sizes for each client follow the power law. We set the mini batch-size to 10 and learning rate $\eta = 0.01$. We report training loss as well as the mean and variance of test accuracies versus the number of communication rounds in Figure 1 for the synthetic IID setting. We observe three key benefits of DivFL compared to random sampling and power-of-choice approaches. On the one hand, DivFL achieves a significant convergence speedup ($\sim 10\times$ faster) to reach the same loss and accuracy relative to random sampling and power-of-choice. The convergence speed could potentially be attributed to the reduction in the variance of updates across epochs as DivFL aims to minimize the gradient approximation error w.r.t the true gradient. Furthermore, DivFL also achieves the lowest loss and highest accuracy among the client selection approaches. By reaching a lower variance of accuracy in comparison to the baselines, DivFL also shows marginal improvement in fairness, even when the data distribution is IID.

As one would expect, while being impractical, utilizing the gradient computation from all clients to update the dissimilarity matrix provides an upper bound on the achievable performance. Our "no overhead" approach to update the dissimilarity matrix partially from only the participating clients still outperforms the baselines in the test accuracy and training loss, while preserving the faster convergence of DivFL. In Appendix B.1, we report the above metrics for different choices of $K$.

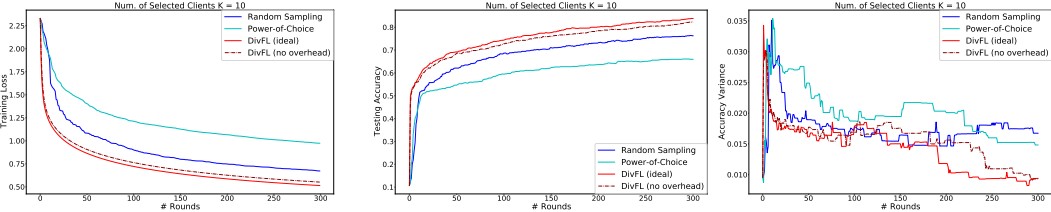

Figure 1: Training loss, mean and variance of test accuracies of DivFL compared with random sampling and power-of-choice on the synthetic IID data. For DivFL (no overhead), we utilize only the gradients from clients participating in the previous round. We see that DivFL achieves faster convergence and converges to more accurate and slightly more fair solutions than all baselines.

We report the training loss as well as the mean and variance of testing accuracies for the synthetic non-IID dataset in Figure 2. We notice the noisy updates of the random sampling approach and slightly less noisy ones of the power-of-choice approach. In addition, both the baselines converge to a less optimal solution than DivFL. The mean accuracy gains of DivFL are quite significant (10% points). In terms of convergence, power-of-choice approach converges in $2x$ fewer rounds than random sampling to reach an accuracy of 70% but DivFL converges in $5x$ fewer rounds with less noisy updates than both baselines. The fairness gains of DivFL for the non-IID setting is more significant. This is due to the higher degree of heterogeneity in client updates and the ability of DivFL to find diverse representative clients in the update space. In Appendix B.1, we provide more ablation studies for different choices of the hyperparameter $K$.

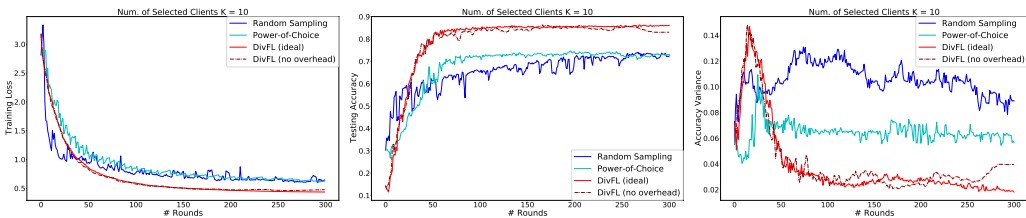

Figure 2: Training loss and test accuracy of DivFL compared with random sampling and power-of-choice on synthetic non-IID data. DivFL converges faster and to more accurate solutions and much improved fairness than all baselines.

## 5.2 RESULTS ON REAL DATASETS

We present extensive results on the robustness of `DivFL` on LEAF federated learning datasets. This includes image datasets (FEMNIST, CelebA) and a language dataset (Shakespeare). For each of these cases, we report the convergence behavior and fairness performance of `DivFL`.

For *FEMNIST*, there are a total of 500 clients where each client contains 3 out of 10 lowercase hand-written characters. Clients use a CNN-based 10-class classifier model with two 5x5-convolutional and 2x2-maxpooling (with a stride of 2) layers followed by a dense layer with 128 activations. For our experiments, *CelebA* contains 515 clients where each client is a celebrity and a CNN-based binary classifier is utilized with 4 3x3-convolutional and 2x2-maxpooling layers followed by a dense layer. For *Shakespeare*, a two-layer LSTM classifier containing 100 hidden units with an 8D embedding layer is utilized. The task is next-character prediction with 80 classes of characters in total. There are a total of 109 clients. The model takes as input a sequence of 80 characters, embeds the characters into a learned 8-dimensional space and outputs one character per training sample after 2 LSTM layers and a densely-connected layer. For the Shakespeare case, each client performs 5 local updates in every global round. For the other two datasets, client updates are shared after only 1 local update.

### 5.2.1 CONVERGENCE BEHAVIOR

We first present the convergence behavior of `DivFL` in comparison to the baselines for the above 3 datasets in Figure 3. On FEMNIST, `DivFL` converges faster than both random sampling and power-of-choice approaches also with less noisy updates. `DivFL` with "no overhead" performs as well as the ideal case. We note that in the case of FEMNIST, there are several clients that have similar distribution over class labels. `DivFL` can be crucial in such cases where redundancies can be minimized and diversity encouraged by maximizing a submodular function.

On CelebA, `DivFL` has a faster convergence than both baselines in the ideal setting. However, the "no overhead" setting converges at the same rate as random sampling followed by the power-of-choice approach. In the case of Shakespeare, interestingly, both `DivFL` and power-of-choice approaches converge at the same rate and faster than random sampling (by 6x). Overall, for CelebA and Shakespeare datasets, we note that `DivFL` either converges at least at the same rate as the fastest baseline.

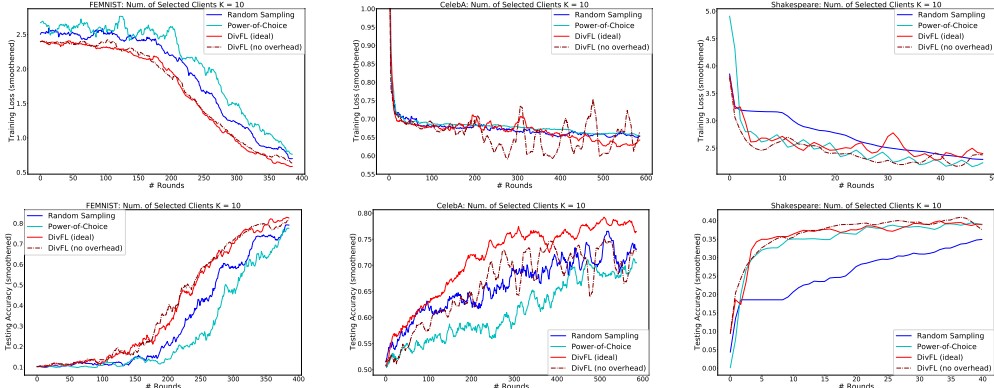

Figure 3: Training loss and test accuracy of `DivFL` compared with random sampling and power-of-choice on real datasets. We observe clear improvement for `DivFL` on FEMNIST. For the other two datasets, the communication-efficient `DivFL` "no-overhead" converges at the same rate as the fastest baseline.

### 5.2.2 FAIRNESS

We also compare the fairness performance of the different approaches for the above 3 real datasets in Figure 4. We measure this through the variance of test accuracies vs training rounds and observe both the trajectory as well as the final variance of the test accuracies. For the image datasets FEMNIST and CelebA, it's clear that `DivFL` has better fairness after convergence with superior mean accuracy and better/comparable variance of accuracies. This shows that diversity helps result in a more uniform performance across clients. Interestingly for the case of Shakespeare, we observe the variance of

accuracies for `DivFL`, especially under the "no overhead" setting is larger than both baselines. This could possibly be the result of the sparsity in the gradient updates for language models, in general. While the random sampling approach has lower accuracy variance, the mean accuracies are also lower showing that it is not necessarily more "fair".

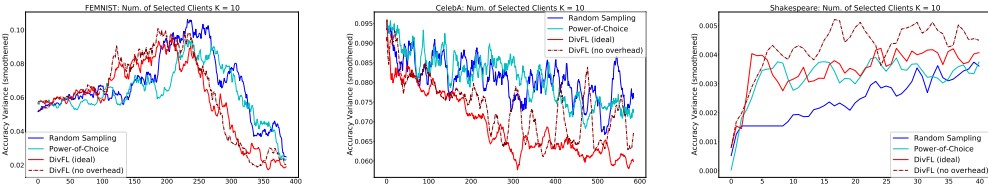

Figure 4: Variance of test accuracies over the 3 real datasets shows that `DivFL` has fairness benefits on the image datasets. Poorer fairness performance for Shakespeare could be attributed to the sparsity in the gradient updates for language models.

### 5.2.3 IMPACT OF K AND LEARNING EFFICIENCY

We also empirically evaluate how the choice of number of clients per round (K) affects the final model performance. We take the case of CelebA dataset and observe the (smoothened) mean and variance of final test accuracies after 400 rounds for different choices of K. The final mean test accuracy generally increases and then decreases for random sampling showing that more participation has the potential to improve the final model performance but not always due to potential risk of overfitting. We also note that for `DivFL`, the test accuracy improves as $K$ grows to 30, but does not fall off as steeply as in the case of random sampling. On the one hand, this highlights the robustness of `DivFL`. On the other hand, `DivFL` achieves the highest mean accuracy of about $0.79$ for $K = 30$. This shows considerably fewer client participation in comparison to $K = 60$ for random sampling and $K = 80$ for power-of-choice to achieve their respective highest test accuracies. This reveals that `DivFL` offers improved learning efficiency by its ability to learn from fewer clients per round compared to the baselines. It must also be noted that `DivFL` outperforms both baselines in final mean and variance of test accuracies for all choices of $K$ in Figure 5.

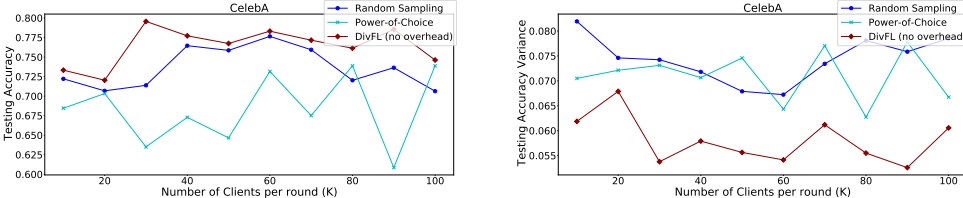

Figure 5: Final Testing Accuracies (smoothened) after 400 epochs on CelebA dataset shows the power of `DivFL` to have robust learning beating baselines for all choices of K. Furthermore, `DivFL` can achieve the highest test accuracy and lowest variance using the smallest number of clients per round in comparison to baselines.

## 6 CONCLUSION

In this paper, we posed the problem of selecting the most diverse subset of clients for federated learning as the solution to maximizing a submodular facility location function defined over gradient space. To this end, we developed `DivFL` and provided a thorough analysis of its convergence in heterogeneous settings. Through extensive empirical results on both synthetic and real datasets, we demonstrated strong gains of `DivFL` over state-of-the-art baseline client selection strategies across key metrics including convergence speed, fairness, and learning efficiency. Furthermore, our communication-efficient variant of `DivFL` holds all these advantages while making them amenable to practical federated learning systems.

## 7 ETHICS AND REPRODUCIBILITY STATEMENT

Our algorithm to select the most diverse set of clients not only aims to speed up learning but can also allow for a fair representation of clients in the training process. This issue has been identified as a key challenge in federated learning in several studies and algorithms exist that specifically address this concern. In our paper, our approach is unique in addressing client diversity (without any additional knowledge of client data) by encouraging diversity in the gradient space with (near) optimality guarantees through the optimization framework.

In order to make our paper reproducible, we provide a detailed proof of Theorem 1 in Appendix A. Further, detailed experimental description including 1) data preparation, 2) model architecture, 3) training algorithm and hyper-parameters are provided in Section 5 and Appendix B.

## ACKNOWLEDGEMENTS

This work was supported in part by the CONIX Research Center, one of six centers in JUMP, a Semiconductor Research Corporation (SRC) program sponsored by DARPA.

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

## APPENDIX

## A COMPLETE CONVERGENCE ANALYSIS

Below shows the convergence analysis. Many steps make a distinction between if we are doing an aggregating steps (from the clients to the server), or not (when the clients do not communicate). We assume that we aggregate every $E$ time steps. Define virtual sequences $\{v_t^k\}_{k \in [N]}$ and $\{w_t^k\}_{k \in [N]}$ where for all $k \in [N]$,

$$v_{t+1}^k = w_t^k - \eta_t \nabla F_k(w_t^k) \tag{8}$$

$$w_{t+1}^k = \begin{cases} v_{t+1}^k & \text{if not aggregating,} \\ \text{sample } S_{t+1} \text{ and average } \{v_{t+1}^k\}_{k \in S_{t+1}} & \text{otherwise.} \end{cases} \tag{9}$$

Let

$$\overline{v}_t := \sum_{k \in [N]} p_k v_k^t, \tag{10}$$

$$\overline{w}_t := \sum_{k \in [N]} p_k w_t^k. \tag{11}$$

where $p_k \geq 0$ is the given weight of the $k^{\text{th}}$ client and w.l.o.g., we assume $\sum_k p_k = 1$. Therefore,

$$\overline{w}_t = \begin{cases} \overline{v}_t & \text{if not aggregating, i.e., when } t \neq \ell E \text{ for some integer } \ell, \\ \frac{1}{K} \sum_{l \in S_t} v_t^l & \text{otherwise.} \end{cases} \tag{12}$$

Let

$$g_t := \sum_{k \in [N]} p_k F_k(w_t^k; \zeta_t^k), \tag{13}$$

and

$$\overline{v}_{t+1} = \overline{w}_t - \eta_t \left( \sum_{k \in [N]} p_k F_k(w_t^k, \zeta_t^k) \right) := \overline{w}_t - \eta_t g_t. \tag{14}$$

We have

$$\|\overline{w}_{t+1} - w^*\|^2 = \|\overline{w}_{t+1} - \overline{v}_{t+1} + \overline{v}_{t+1} - w^*\|^2 \tag{15}$$

$$= \|\overline{w}_{t+1} - \overline{v}_{t+1}\|^2 + \|\overline{v}_{t+1} - w^*\|^2 + 2\langle \overline{w}_{t+1} - \overline{v}_{t+1}, \overline{v}_{t+1} - w^* \rangle. \tag{16}$$

If not aggregating,

$$\overline{w}_{t+1} = \overline{v}_{t+1}. \tag{17}$$

Hence

$$\|\overline{w}_{t+1} - w^*\|^2 = \|\overline{v}_{t+1} - w^*\|^2. \tag{18}$$

Using Lemma 1 in Li et al. (2019), we know $\mathbb{E}[\|\overline{v}_{t+1} - w^*\|^2] \leq (1 - \eta_t \mu)\mathbb{E}[\|\overline{w}_t - w^*\|^2] + \eta_t^2 B$ holds for some constant $B$. If we are aggregating, we need to bound

$$\mathbb{E}[\|\overline{w}_{t+1} - \overline{v}_{t+1}\|^2] + \mathbb{E}[\|\overline{v}_{t+1} - w^*\|^2] + 2\mathbb{E}[\langle \overline{w}_{t+1} - \overline{v}_{t+1}, \overline{v}_{t+1} - w^* \rangle]. \tag{19}$$

Let the last time of aggregation happens at step $t_0 = t + 1 - E$, when we select a subset $S$ (associated with weights $\{\gamma_k\}_{k \in S}$) using the greedy algorithm. Let $\Delta v_\tau^k$ be the updates on $v^k$ at the $\tau$-th iteration, i.e., $\Delta v_\tau^k := v_{\tau+1}^k - v_\tau^k$. Then $\overline{v}_{t+1} = \overline{w}_{t_0} + \frac{1}{N} \sum_{k \in [N]} \sum_{\tau=t_0}^t \Delta v_\tau^k$. To bound the first term above,

$$\|\overline{w}_{t+1} - \overline{v}_{t+1}\| = \left\| \left( \overline{w}_{t_0} + \frac{1}{N} \sum_{k \in S} \gamma_k \sum_{\tau=t_0}^t \Delta v_\tau^k \right) - \left( \overline{w}_{t_0} + \frac{1}{N} \sum_{k \in [N]} \sum_{\tau=t_0}^t \Delta v_\tau^k \right) \right\| \tag{20}$$

$$= \left\| \sum_{\tau=t_0}^t \left( \frac{1}{N} \sum_{k \in S} \gamma_k \Delta v_\tau^k - \frac{1}{N} \sum_{k \in [N]} \Delta v_\tau^k \right) \right\| \tag{21}$$

$$\leq \sum_{\tau=t_0}^t \left\| \frac{1}{N} \sum_{k \in S} \gamma_k \Delta v_\tau^k - \frac{1}{N} \sum_{k \in [N]} \Delta v_\tau^k \right\| \tag{22}$$

Similar to the CRAIG paper (Mirzasoleiman et al., 2020), we assume that the subset $S$ selected in step $t_0 = t + 1 - E$ provides an approximation of the full gradient such that

$$\left\| \frac{1}{N} \sum_{k \in S} \gamma_k \nabla F_k(v_{t_0}^k) - \frac{1}{N} \sum_{k \in [N]} \nabla F_k(v_{t_0}^k) \right\| \leq \epsilon, \tag{23}$$

For every local step $\tau \in (t_0, t]$, we use the same $S$ to approximate the full gradient because we only communicate the local gradients every $E$ local steps. To bound the gradient approximation at step $\tau$ using the stale $S$, we have

$$\left\| \frac{1}{N} \sum_{k \in S} \gamma_k \nabla F_k(v_\tau^k) - \frac{1}{N} \sum_{k \in [N]} \nabla F_k(v_\tau^k) \right\| \leq \left\| \frac{1}{N} \sum_{k \in S} \gamma_k \nabla F_k(v_\tau^k) - \frac{1}{N} \sum_{k \in S} \gamma_k \nabla F_k(v_{t_0}^k) \right\| + \tag{24}$$

$$\left\| \frac{1}{N} \sum_{k \in S} \gamma_k \nabla F_k(v_{t_0}^k) - \frac{1}{N} \sum_{k \in [N]} \nabla F_k(v_{t_0}^k) \right\| + \tag{25}$$

$$\left\| \frac{1}{N} \sum_{k \in [N]} \nabla F_k(v_\tau^k) - \frac{1}{N} \sum_{k \in [N]} \nabla F_k(v_{t_0}^k) \right\| \tag{26}$$

$$\leq 2LG \sum_{\nu = t_0}^{\tau} \eta_\nu + \epsilon, \tag{27}$$

where the first and the third term on the right hand side are bounded using the $L$-smoothness of $F_k(\cdot)$ and $G$-bounded norm of its stochastic gradient. Hence, we can continue to bound the first term in Eq. (22) by

$$\|\overline{w}_{t+1} - \overline{v}_{t+1}\| \leq \sum_{\tau = t_0}^{t} \left\| \frac{1}{N} \sum_{k \in S} \gamma_k \Delta v_\tau^k - \frac{1}{N} \sum_{k \in [N]} \Delta v_\tau^k \right\| \tag{28}$$

$$= \sum_{\tau = t_0}^{t} \eta_\tau \left\| \frac{1}{N} \sum_{k \in S} \gamma_k \nabla F_k(v_\tau^k) - \frac{1}{N} \sum_{k \in [N]} \nabla F_k(v_\tau^k) \right\| \tag{29}$$

$$\leq 2LG \sum_{\tau = t_0}^{t} \sum_{\nu = t_0}^{\tau} \eta_\tau \eta_\nu + E\epsilon \eta_\tau \tag{30}$$

$$\leq LGE(E-1)\eta_{t_0}^2 + E\epsilon \eta_{t_0} \tag{31}$$

$$= LGE(E-1) \left( 1 + \frac{E-1}{t + \gamma - (E-1)} \right)^2 \eta_t^2 + E\epsilon \left( 1 + \frac{E-1}{t + \gamma - (E-1)} \right) \eta_t \tag{32}$$

where $E$ is the number of local steps between two communication (aggregation) rounds. Therefore, Eq. (19) can be bounded as follows:

$$\mathbb{E}[\|\overline{w}_{t+1} - w^*\|^2] \tag{33}$$

$$\leq \mathbb{E}[\|\overline{w}_{t+1} - \overline{v}_{t+1}\|^2] + \mathbb{E}[\|\overline{v}_{t+1} - w^*\|^2] + 2\mathbb{E}[\langle \overline{w}_{t+1} - \overline{v}_{t+1}, \overline{v}_{t+1} - w^* \rangle] \tag{34}$$

$$\leq \left( LGE(E-1)\eta_{t_0}^2 + E\epsilon \eta_{t_0} \right)^2 + \left[ (1 - \eta_t \mu)\mathbb{E}[\|\overline{w}_t - w^*\|^2] + \eta_t^2 B \right] + 2 \left( LGE(E-1)\eta_{t_0}^2 + E\epsilon \eta_{t_0} \right) \mathbb{E}[\|\overline{v}_{t+1} - w^*\|] \tag{35}$$

$$\leq (1 - \eta_t \mu)\mathbb{E}[\|\overline{w}_t - w^*\|^2] + E\epsilon \rho \eta_{t_0} + \left[ LGE(E-1)\rho + (LGE(E-1)\eta_{t_0} + E\epsilon)^2 \right] \eta_{t_0}^2 + B\eta_t^2 \tag{36}$$

$$\leq (1 - \eta_t \mu)\mathbb{E}[\|\overline{w}_t - w^*\|^2] + \epsilon \rho E \eta_{t_0} + \left[ LG\rho + (LGE\eta_{t_0} + \epsilon)^2 \right] E^2 \eta_{t_0}^2 + B\eta_t^2, \tag{37}$$

where $\eta_t = \frac{\beta}{t+\gamma}$, $\eta_{t_0} = \frac{\beta}{t-E+1+\gamma}$ and $\mathbb{E}[\|\overline{v}_{t+1} - w^*\|] \leq \rho$, shown as follows.

$$\mathbb{E}\left\|\overline{v}^{t+1} - w^*\right\| \leq \mathbb{E}\left\|\overline{v}^{t+1} - \sum_{i \in [N]} p_i v_i^*\right\| + \mathbb{E}\left\|\sum_{i \in [N]} p_i v_i^* - w^*\right\| \tag{38}$$

$$\leq \mathbb{E}\left\|\overline{v}^{t+1} - \sum_{i \in [N]} p_i v_i^*\right\| + M \tag{39}$$

$$\leq \sum_{i \in [N]} \mathbb{E}\|p_i(\overline{v}_i^{t+1} - v_i^*)\| + M \tag{40}$$

$$\leq \sum_{i \in [N]} \frac{p_i}{\mu} \mathbb{E}\left\|\nabla f_i(v_i^t)\right\| + M \tag{41}$$

$$\leq \frac{G}{\mu} + M \leq \rho. \tag{42}$$

The final convergence rates follows from Lemma 3 in Mirzasoleiman et al. (2020).

## B  ADDITIONAL EXPERIMENTS

### B.1  MORE ABLATION STUDIES ON SYNTHETIC AND FEMNIST DATASETS

We present more findings from training $\mathtt{DivFL}$ on synthetic IID and non-IID dataset for $K = 20$ in Figure 6, 7 and 8. In addition to the training loss, mean and variance of test accuracies, we also plot the 10th percentile test accuracy that shows the worst-case performance of the different client selection strategies. As in the case of $K = 10$, we observe gains across convergence, fairness and improved model accuracy for $\mathtt{DivFL}$. In addition, the 10th percentile accuracy further confirms that $\mathtt{DivFL}$ improves the worst-case clients' accuracies.

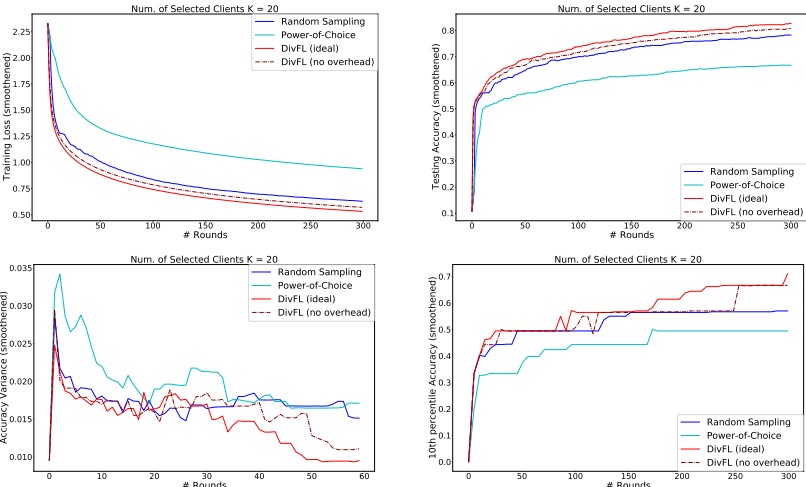

Figure 6: We measure the training loss, mean and variance of test accuracies as well as 10th percentile test accuracy for $\mathtt{DivFL}$ on synthetic IID dataset for $K = 20$.

### B.2  IMPACT OF NUMBER OF LOCAL EPOCHS $\tau$

In our experiments, we already adopted local updating schemes with a fixed number of local epochs ($\tau$) being 1 (running multiple local iterations). However, we further show the robustness of $\mathtt{DivFL}$ under different choices of $\tau$ via observing the training loss and variance of test accuracies. The results for Synthetic IID, non-IID and FEMNIST are presented in Figure 9, 10, 11, 12, 13 for different values of $\tau$. We observe that for different choices of the number of clients $K$ and the number of local epochs

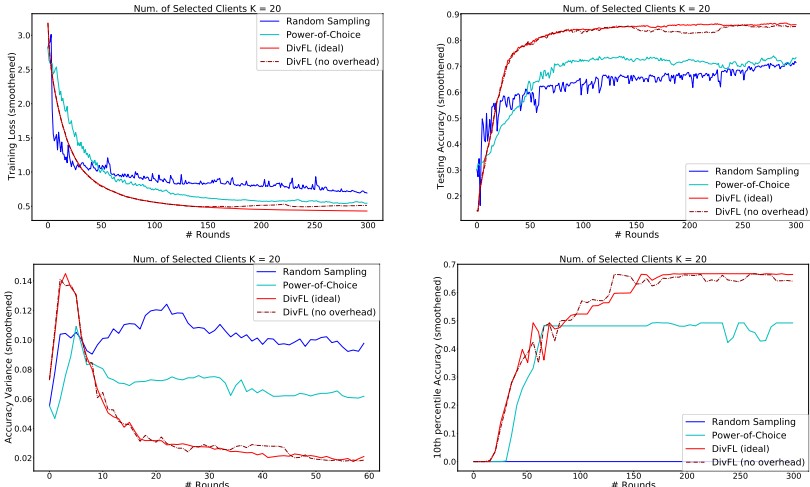

Figure 7: The training loss, mean and variance of test accuracies as well as 10th percentile test accuracy for $\mathtt{DivFL}$ on synthetic non-IID dataset for $K = 20$.

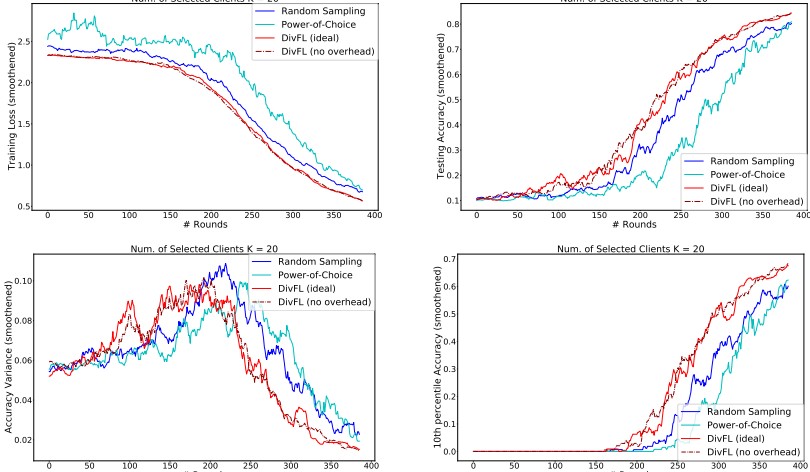

Figure 8: Training loss, mean and variance of test accuracies as well as 10th percentile test accuracy for $\mathtt{DivFL}$ on FEMNIST dataset for $K = 20$.

$\tau$, both variants of DivFL converge faster than both the baselines. The fairness gains over random sampling are also preserved under large values of $\tau$ as shown in Figure 13.

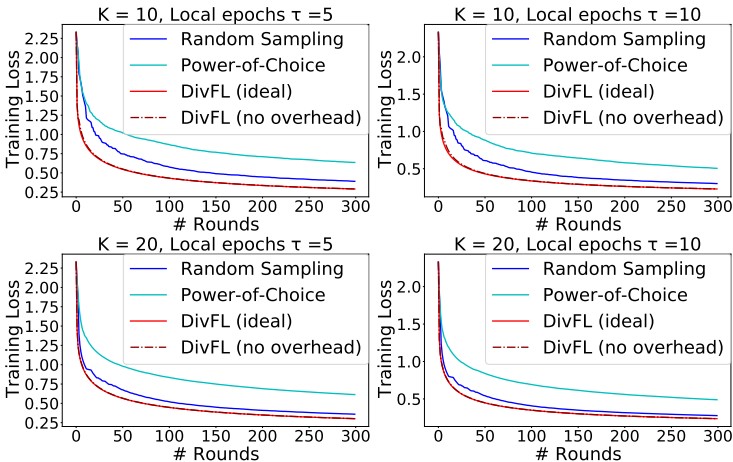

Figure 9: Training Loss on Synthetic IID for different choices of $\tau$.

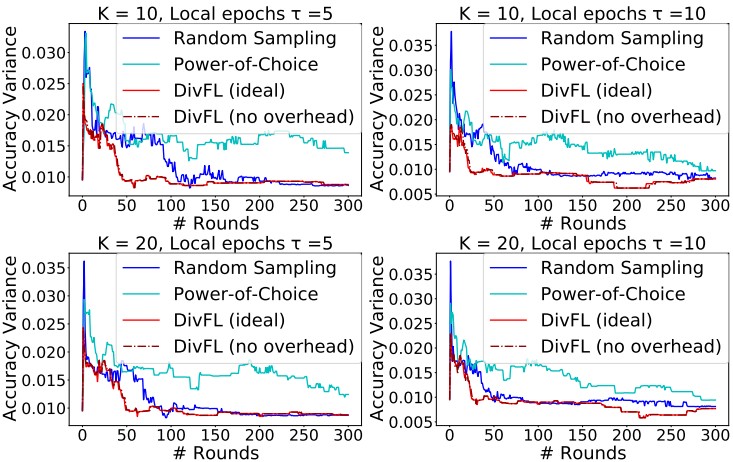

Figure 10: Variance of Test Accuracy on Synthetic IID

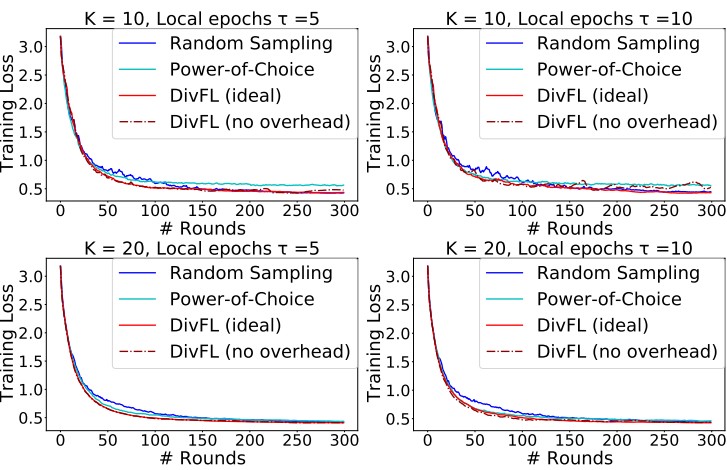

Figure 11: Training Loss on Synthetic non-IID for different choices of $\tau$.

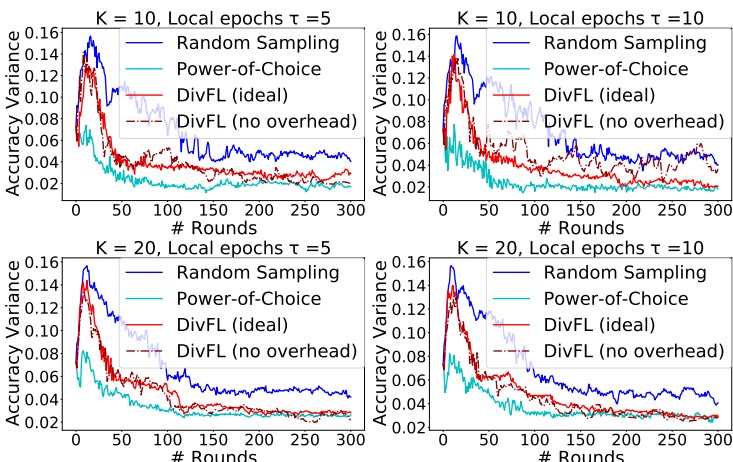

Figure 12: Variance of Test Accuracy on Synthetic non-IID

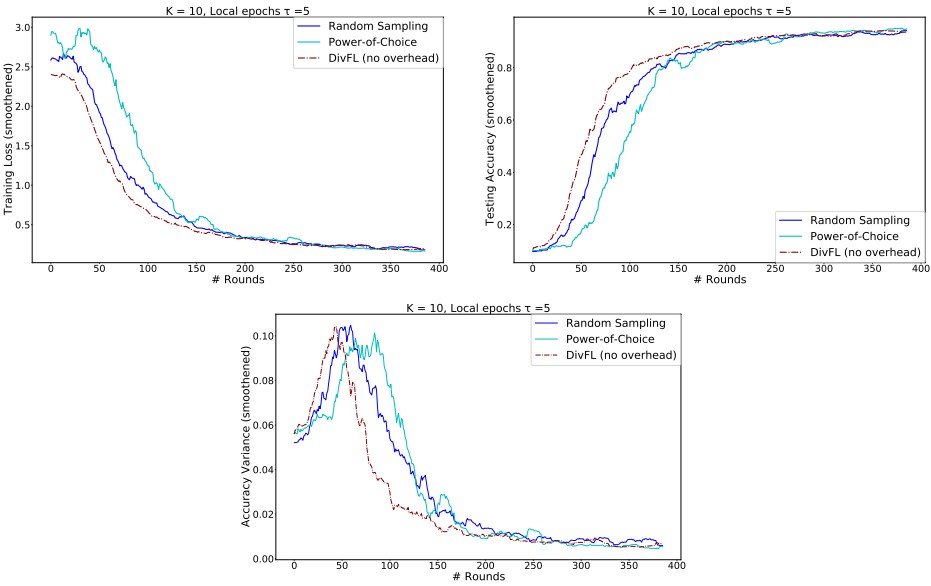

Figure 13: Faster convergence benefits of `DivFL` over Random Sampling and Power-of-Choice approaches is preserved even when clients run multiple local epochs before sharing model updates.

