# OpenReview forum: "Diverse Client Selection for Federated Learning via Submodular Maximization"
_ICLR.cc/2022/Conference — ICLR 2022 Poster_

### Official Review · Reviewer_CFhn · 2021-10-27

**Correctness:** 3
**Technical Novelty And Significance:** 3
**Empirical Novelty And Significance:** 3
**Recommendation:** 8
**Confidence:** 3

**Main Review:**

### Strengths

- Overall, the paper is very well written. The proposed work is well-motivated, as developing better client sampling techniques is one important direction in FL.
- Proposed approach is technically solid and seems novel enough. Although formulating client selection in FL by a combinatorial optimization problem is not completely new (e.g., Nishio and Yonetani 2019), the objective of approximating the aggregation of gradients of all clients by that from a client subset is novel, as far as I can check. I think that this is a reasonable objective for both data iid and non-iid settings and indeed worked well as shown in the experiments.
- With some reasonable assumptions, the convergence analysis for the proposed approach is provided.

### Weaknesses
Although the proposed approach is technically sound, requiring the local gradients of all clients becomes a critical limitation to adopt the approach to practical problems, as also pointed out in the manuscript. Utilizing updates collected in previous rounds would be one possible approach, but it was not fully evaluated in the experiments in my opinion.

Specifically, using previous gradients will inevitably come with a stale gradient problem (as also noted in the paper). If I understand it correctly, this problem was mitigated in the experiments by limiting the number of local updates to be one for each client so as not to make model weights change dramatically over rounds. However, this is against a previous finding of federated learning where asking each client to update models multiple times leads to efficient training, as confirmed in FedAvg (McMahan et al, 2017). Even though limiting the number of local epochs could result in no significant loss when using previous gradients, does it make the overall learning procedure inefficient compared to when clients perform multiple local updates before aggregation? This is not obvious in the current manuscript. To address this concern, I'd like to see how training curves and final performances change depending on the number of local updates (\tau parameter in the experiment).

### Minor comments
- The text size of the plot labels and ticks should be a bit larger.
- The left margin of Algorithm 1 needs to be a bit larger.

**Summary Of The Paper:**

This paper presents a new client selection strategy for federated learning. The key idea is to view the client selection problem as a facility location problem, where the objective is to approximate the aggregation of gradients of all the clients with that of a subset of selected clients. A greedy solution to the problem is derived, and the theoretical convergence analysis is also provided. Although the proposed algorithm ideally requires the gradients of all clients, the paper also introduces its practical extension that instead leverages gradients collected in previous rounds. Experimental results on synthesized and real datasets show the effectiveness of the proposed approach.

**Summary Of The Review:**

Overall, I think this is a nice paper that addresses an important aspect of federated learning: improving a client sampling strategy. Experimental results are promising, though I would like to see how not limiting the number of local updates will affect the performances. If this brings a considerable gap between DivFL (ideal) and DivFL (no overhead), that implies the limitation of the proposed approach for practical settings.

---

> ### Author Response · Authors · 2021-11-18
> **Response to reviewer CFhn**
>
> We thank the reviewer for their positive and insightful review.
>
> **[Impact of # local epochs]** The reviewer is correct that the performance of DivFL will be affected by the number of local epochs, as the estimation of the device similarity matrix relies on stale gradients from previously selected devices. In the initial submission, we already adopted local updating schemes with a fixed number of local epochs ($\tau$)  being 1 (running multiple local iterations).  To further address the reviewer’s concern, we added additional experiments showing the test performance v.s. the number of epochs. The results are shown for Synthetic and FEMNIST datasets in Appendix B.2 of the updated submission. We observe that for different choices of the number of clients $K$ and the number of local epochs $\tau$, both variants of DivFL converge faster than both the baselines. The fairness gains over random sampling are also preserved under large values of $\tau$.
>
> **[Others]** Thank you for the suggestions. We have adjusted the left margin of Algorithm 1, and will adjust the text/tick size of figures in the updated version.

---

> > ### Comment · Reviewer_CFhn · 2021-11-23
> > **Thanks**
> >
> > Thank you very much for the response. I somehow mistook the number of local epochs for the number of local updates. Nevertheless, the results of additional experiments provided in the updated manuscript are valuable. I'll keep my original rating.

---

### Official Review · Reviewer_2aPT · 2021-11-02

**Correctness:** 4
**Technical Novelty And Significance:** 2
**Empirical Novelty And Significance:** 4
**Recommendation:** 6
**Confidence:** 2

**Main Review:**

The paper is overall well-written and somehow easy to follow (see minor comments). I checked most of the math in the Appendix and it seems correct, but standard. The authors could emphasize more on the key differences between their analysis and the previous literature.

Main questions:
-	I understand the use of the standard greedy algorithm for the selection of clients in DivFL, however, the role of submodularity doesn’t seem to give an advantage, at least theoretically speaking. In other words, which role does submodularity play in the analysis of the algorithm? Is there any connection between Assumption 1 and submodularity?
-	Assumption 1 seems rather strong. The authors should motivate more this. For example, which classes of functions satisfy this? How large is $\epsilon$ in that case or in the worst-case?
-	The authors focus on the full communication case, can this approach be adapted to the partial case?
-	Experiments: the authors should discuss Assumption 1 for the objectives used in this section, at least to have a sense of how large is $\epsilon$ for these applications. Also, it would be great to have an idea of the of time that each method spends.

Minor comments:
-	Assumption 4 and 5 are for all t?
-	Page 5: in the definition of $\overline{v}_t$, it is $v^k_t$.
-	The Setup paragraph in Page 5 is a bit confusing, it would be great if the authors could explain more the role of $\overline{v}$ and $\overline{w}$, in particular, why they are needed for the analysis in Theorem 1.
-	Appendix: $g_t$ and $\Delta v^k_t$ are not defined.



**Summary Of The Paper:**

This paper studies the selection of clients for federated learning under the assumption of full participation via submodular function maximization. Specifically, the goal is to obtain a subset of the clients such that the aggregation of the gradients of their loss functions approximates the full aggregated gradient. The authors show that the error defined by the difference between both aggregations (the full one and the one defined by the subset) can be upper bounded by a supermodular function. Therefore, minimizing this function (subject to a single cardinality constraint, which is motivated by communication costs) gives an upper bound for the problem of minimizing the error. The minimization of this particular supermodular function can be equivalently posed as the maximization of a submodular function, which has been previously studied in the literature (e.g. [1] below). Given this, the authors propose a variant of the federated averaging scheme, called *DivFL*, which uses the standard greedy algorithm to select $K$ clients to communicate the current value $w_t$. They show that under the “gradient approximation error” assumption, the error between $w_t$ and $w^*$ decreases as $O(1/t)+O(\epsilon)$, where $\epsilon$ is a parameter of the assumption. Finally, they test their scheme in synthetic and real-data sets against other methods in the literature.


[1] Ryan Gomes and Andreas Krause. Budgeted Nonparametric Learning from Data Streams, 2015.

**Summary Of The Review:**

The paper is interesting and well-written. The convergence analysis seems to be correct. However, the theoretical role of submodularity is not quite clear, besides the use of the greedy algorithm for the selection of clients. Assumption 1 seems quite strong, the authors should motivate this part more in detail. The experiments clearly show the advantage of DivFL over previous approaches in the literature.

---

> ### Author Response · Authors · 2021-11-18
> **Response to reviewer 2aPT**
>
> **[Role of submodularity in the analysis]** Minimizing the upper bound of the approximation error is equivalent to minimizing $G(S)$ (a supermodular minimization problem) (Eq (4)), which is solved by running the greedy selection step in Algorithm 1. In other words, our submodularity-based algorithm effectively minimizes the epsilon value in Assumption 1 (see our response to all reviewers for more details). The baseline of random sampling has no guarantee of approximating the solution that minimizes the error. In contrast, DivFL leads to a more accurate solution (due to minimizing $\epsilon$ compared to other client selection approaches) and also a more representative set of participating clients.
>
> **[Values of $\epsilon$]** Related to the point above, the proposed algorithm is effectively minimizing $\epsilon$. This is not a strong assumption; please see our response to all reviewers for more detail as well as the exact values of $\epsilon$ compared with those of random sampling.
>
> **[Full v.s. partial communication]** Our algorithm in fact already allows for partial device participation (see algorithm description before Section 4), and we use partial participation in our experiments (Section 5). At each communication round, we run a stochastic version of the greedy algorithm by first randomly sampling a small number of clients, and then selecting diverse ones based on the current similarity matrix. We also update the dissimilarity matrix using gradients from only selected clients, and update the global model parameter using model updates from the same set of clients. In our convergence analysis, we also assume only a subset of clients are selected at each actual communication round. We have clarified this in Section 3.2 (greedy selection of clients) and Section 5 (experiments).
>
> **[Experiments]** (a) $\epsilon$ values: please see our responses to all reviewers above for a comparison of $\epsilon$ for random sampling and DivFL showing that DivFL reduces the epsilon and how it varies with $K$.  (b) Since DivFL does not need additional communication to estimate the similarity matrix, the major overhead is running the greedy selection algorithm at the server side, which is negligible if the server is not resource-constrained. Therefore, DivFL roughly takes the same computation time as FedAvg with random sampling.
>
> **[Others]** (a) Yes, Assumption 4 and 5 are for all $t$. (b) We have fixed the typos the reviewer mentioned, clarified a couple notations, and added more explanation on the virtual sequences in the updated submission.

---

### Official Review · Reviewer_RXZV · 2021-11-04

**Correctness:** 2
**Technical Novelty And Significance:** 2
**Empirical Novelty And Significance:** 2
**Recommendation:** 3
**Confidence:** 4

**Main Review:**

page 4: Minimizing G(S) is certainly equivalent to maximizing \bar{G}. However, the equivalence
breaks down when you consider approximation. In particular, the (1-1/e) approximation to \bar{G}
does not imply a similar approximation to G(S), in general

End of section 3: the greedy step is not clearly explained and needs more discussion, since it will
affect the communication complexity. It is mentioned that the algorithm only utilizes
"updates from the clients that participated in the previous round", and it is claimed that this works
quite well. This is pretty surprising. Is there some special structure in the instance due
to which this holds?

Assumption 1: this is a property of the algorithm. So I don't understand why this is an assumption.
You need to prove that for the sets S_t chosen in each step t, this property will hold. The authors
claim that this assumption is also made in (Mirzasoleiman et al., 2020). They should say more clearly
where that is (though an assumption in a prior paper doesn't automatically make it correct).

In view of the issue with assumption 1, it is not clear if Theorem 1 is correct

The formulation in the paper has a lot of similarity with the setup in (Mirzasoleiman et al., 2020).
The authors should explain this more carefully, instead of cursorily stating it in the Introduction

**Summary Of The Paper:**

The paper studies a problem of improving the efficiency of federated learning by selecting a
subset of clients whose gradients are representative. This is formulated as a submodular maximization
problem, and the authors analyze a greedy algorithm for it. This is evaluated through experiments
on synthetic as well as real datasets.

**Summary Of The Review:**

The problem considered in the paper is interesting and well motivated. However, the formulation is
incremental, and builds quite heavily on earlier paper. The main contribution is the analysis of the
greedy algorithm, which has serious issues as mentioned above.

---

> ### Author Response · Authors · 2021-11-18
> **Response to reviewer RXZV**
>
> **[Approximation of $G(S)$]**
> The reviewer is correct that the transformation does not preserve a $(1-\frac{1}{e})$ approximation to $G(S)$, but it still preserves an additive and multiplicative approximation to $G(S)$. This transformation (from a min-form of supermodular function to a max form of a monotone submodular facility location function) is quite straightforward and is standard in prior work on submodular optimization (see, e.g., [1, 2]). We provide a simple proof explaining the additive and multiplicative approximation we can get from this transformation below.
>
> We define $\bar G(S) = c - G(S)$ for a suitable small constant c, for e.g., $c\geq \min_{s\in V} G(\{s\})$. The constant is data dependent but can be made small depending on the constraints. Greedy algorithm applied to the submodular function $\bar G$ guarantees $\bar G(S)\geq (1-e^{-1})\bar{G}(S^*)$, so we have $G(S)\leq G(1-e^{-1})G(S^*)+ce^{-1}$, which provides an upper bound for the approximation error.
>
> The proof is as follows (for the unconstrained case, but the constrained case is similar). We have
> $$ \bar{G}(S) \geq \left(1-\frac{1}{e}\right) \bar{G}(S*)  = \left(1-\frac{1}{e}\right) \max_{S\subseteq V } \bar{G}(S) = \left(1-\frac{1}{e}\right) \max_{S\subseteq V } [c - G(S)]  = \left(1-\frac{1}{e}\right) [c - \min_{S\subseteq V } G(S)] = \left(1-\frac{1}{e}\right)c - \left(1-\frac{1}{e}\right) \min_{S\subseteq V } G(S). $$
> Hence,
> $$  c-G(S)  \geq Constant - \left(1-\frac{1}{e}\right) \min_{S\subseteq V } G(S),  $$
> which implies
> $$ G(S) \leq c_1 + \left(1-\frac{1}{e}\right) \min_{S\subseteq V } G(S).$$
>
>
> **[Greedy selection step]** Thanks for this suggestion. We would like to clarify that we run a stochastic version of the greedy algorithm (not the deterministic greedy algorithm) which allows us to incorporate more randomness in the sampling procedure. Given a client similarity matrix and a set of active clients, at each iteration of solving for the submodular maximization problem (defined in Eq. (5)), the stochastic greedy algorithm selects a client $k^*$ from a small random subset of $V\backslash S$. While most of the cells in the similarity matrix are stale, the random sampling process (before diverse selection) is independent across iterations. We have added the equations on the stochastic version along with more detailed explanations throughout the paper.
>
> **[Assumption 1/Convergence analysis]** Thanks for your comments. (a) This assumption appears in Mirzasoleiman et al (2020), Theorem 1. We have added a note on this in the paper. (b) Assumption 1 is valid, as there always exists an upper bound $\epsilon$ for the gradient approximation. However, we agree that this can be viewed more generally as a property of the algorithm (i.e., there always exists an upper bound $\epsilon$ for the gradient approximation). The convergence rate in Theorem 1 depends on O($\epsilon$), so a larger $\epsilon$ leads to a worse bound. Our theorem does not need $\epsilon$ to be smaller than some value to hold. The goal of our algorithm is to effectively minimize an upper bound of epsilon to achieve a potentially small value, as minimizing the approximation can be formulated as a supermodular minimization problem (Eq (4)), which is what we solve in the client selection algorithm.  Please see our response to all reviewers above for a more detailed response.
>
> **[Differences with Mirzasoleiman et al., (2020)]** The reviewer is correct that we mentioned connections with this work in our original submission, e.g., in Section 1 and Section 3. A major difference is that we explore the diverse subset selection idea for the novel application of federated learning with several fundamental challenges that do not exist in centralized settings studied in Mirzasoleiman et al (2020). In particular, our theory and empirical results model the practical scenarios with limited communication and unreliable networks, which are critical constraints in realistic FL applications (McMahan et al., 2017). These issues were not considered in prior work and affected both the method and resulting theoretical guarantees. Please also see our responses to Reviewer KjQr for more details. We have updated the submission to clarify this (Section 1).
>
>
>
>
> [1] Baharan Mirzasoleiman, Jeff Bilmes, and Jure Leskovec.  Coresets for data-efficient training ofmachine learning models. ICML, 2020.
>
> [2] Ryan Gomes and Andreas Krause. Budgeted Nonparametric Learning from Data Streams. ICML, 2010.

---

> > ### Comment · Reviewer_RXZV · 2021-11-29
> > **Response**
> >
> > Thanks for the response to my comments. Regarding the approximation, the transformation you mention is obvious, but it is not clear how large c_1 is. Can you comment on it in your experiments?
> >
> > The bound on \epsilon is not clear from the other review responses, and needs to be clarified better

---

> > > ### Author Response · Authors · 2021-11-30
> > > **Response to your remaining concerns**
> > >
> > > **[C_1]**
> > > - In theory, as we mentioned in our response, $c$ (and $c_1=(1-\frac{1}{e})c$) is data dependent and the theoretical bound holds for $c=\min_{s\in V} G(\{s\})$.
> > > - In experiments, we do not need to set the value of $c_1$ because running a greedy (or stochastic greedy) algorithm to minimize $G(S)$ produces exactly the same solution set as using the algorithm to maximize $c-G(S)$ for any constant $c$. In other words, changing $c_1$ does not change the experimental results.
> > > **[$\epsilon$ values]**
> > > -  As stated in our previous reply, there always exists an upper bound $\epsilon$ for the gradient approximation. The convergence rate in Theorem 1 depends on O($\epsilon$), so a larger $\epsilon$ leads to a worse bound. Our theorem does not need $\epsilon$ to be smaller than some value to hold. In the *‘response to all reviewers’* above, we also provide the exact values of $\epsilon$ compared with those of random sampling in our experiments (see the table). The results demonstrate that $\epsilon$ is sufficiently small to produce promising FL models in DivFL.
> > >
> > > Please let us know if the above address your concerns or not. Thanks!

---

### Official Review · Reviewer_KjQr · 2021-11-08

**Correctness:** 4
**Technical Novelty And Significance:** 3
**Empirical Novelty And Significance:** 3
**Recommendation:** 6
**Confidence:** 3

**Main Review:**

Subdmoularity-based subset selection is widely used in many contexts, but as far as I can tell, it's the first time in federated learning. Use of gradients in this context is not novel.

Theoretical and empirical results are significant.

**Summary Of The Paper:**

This paper studies the use of diversity sampling (based on submodular functions) for selecting clients who send updates to the server in a federated learning environment. As the authors show, this provides more efficient use of communication budget in terms of convergence and model accuracy.

**Summary Of The Review:**

The paper proposes a new approach for diverse sampling of clients in each update round of federated learning. This significantly improves convergence rate and model accuracy. Though the idea is not particularly novel in the broader context, it does advance state of the art in federated learning.

---

> ### Author Response · Authors · 2021-11-18
> **Response to reviewer KjQr**
>
> **[Novelty of DivFL]**  We thank the reviewer for the positive review of the submission. We agree that the high-level idea of diverse subset selection based on submodularity has been extensively studied in various problems. However, it is not straightforward to extend prior work on this problem to application of federated client selection. In federated networks, we perform selection over clients based on model updates (not gradients from individual samples). In addition, we allow for partial device participation where the server does not have access to all data at any communication round, as is standard in FL (McMahan et al., 2017). Our proposed algorithm and convergence analysis are compatible with these constraints. In particular, we make several algorithmic changes to the general subset selection problem including (a) incorporating systems constraints by selecting diverse clients from a randomly-chosen active set via a stochastic version of the greedy algorithm, and (b) estimating the client similarity matrix using partial gradients obtained from selected (not all) clients after running local SGD. Our work advances state-of-the-art by providing a principled method (DivFL) that outperforms client selection baselines in realistic federated settings with minimal overhead.

---

### Author Response · Authors · 2021-11-18
**Response to all reviewers**

We thank the reviewers for their time and valuable feedback, which will improve the submission. We first respond to a shared concern related to Assumption 1 and then address specific comments.

**[Assumption 1, $\epsilon$ values, and convergence analysis]**

* **Correctness**: First, we note that Theorem 1 together with the analysis is correct no matter whether the gradient approximation error $\epsilon$ is large or small. In particular, there always exists an upper bound $\epsilon$ for the gradient approximation. The bound in Theorem 1 depends on O($\epsilon$), so a larger $\epsilon$ leads to a worse bound.

* **Goal of our algorithm**: Our proposed algorithm effectively minimizes the upper bound to achieve a small $\epsilon$ by selecting a subset of clients with greedy algorithms (as explained under Assumption 1). The purpose of Algorithm 1 is to make the theorem more applicable. $\epsilon$ is data-dependent and it is possible to empirically compute an upper bound for it by minimizing the norm of the gradient difference given the selected subset by the greedy algorithm. We have clarified this in the updated submission.
The final experimental results demonstrate that $\epsilon$ is sufficiently small to produce promising FL models. To this end, we computed the mean and standard deviation of $\epsilon$ over the training epochs resulting from DivFL and random sampling based client selection. The dataset utilized was synthetic non-IID and the number of participating clients ($K$) varied to study the impact on the gradient approximation error. The results (unnormalized) are tabulated below. For all choices of K, DivFL results in lower $\epsilon$ as shown by the mean and standard deviation. As expected, $\epsilon$ becomes smaller and smaller as K increases towards full device participation but the mean magnitudes are significantly smaller for DivFL (~2$\times$ smaller especially for $K$>=10).

| # selected clients  (out of 30)  |  [Random Sampling]  $\epsilon$ mean (std) |  [DivFL (ideal)]  $\epsilon$ mean (std) |
|--- |---|---|
|  5 | 1.379 (0.96)  | 1.067 (0.93) |
| 10 | 0.951 (0.56)  |  0.575 (0.578) |
|  15|  0.648 (0.38) |  0.38 (0.383) |
|  20|  0.469 (0.292) | 0.248 (0.245) |
|  25| 0.277 (0.2) |  0.131 (0.12) |

* **Selection**: Finally, we currently perform selection under a cardinality constraint on the clients, which will give some bounded $\epsilon$ values. Alternatively, we can set a small target $\epsilon$ and optimize for a subset without size constraints (thus potentially increasing the number of selected clients).

---

### Decision · Program_Chairs · 2022-01-20

**Decision:**

Accept (Poster)

**Comment:**

The paper proposes a novel method for (diverse) client selection at each round of a federated learning procedure with the aim of improving performance in terms of convergence, learning efficiency and fairness. The main idea is to  introduce a facility location objective to quantify how representative/informative is the gradient information of a given set of clients is, and then choose a subset that maximizes this objective.  Given the monotonicity and submodularity of the proposed facility location objective, the authors have been able to provide theoretical guarantees. Experimental results on two data sets (FEMNIST and CelebA) show the effectiveness to the proposed approach and algorithm.

The reviewers had a number of concerns most of which were addressed in the authors response. The reviewers believe that the theoretical results of the paper are incremental given the prior work (see the reviews for more details); however, the reviewers (as well as myself) agree that the proposed method is novel and can provide significant practical advantage. Utilizing sub modular objectives for diverse selection is a well-known (and effective approach), but I am seeing it in the context of federated learning for the first time.

My suggestion to the authors: (i) Improve the experimental section by adding a few more common data sets (such as CIFAR when data is distributed in a heterogeneous manner). CelebA and FEMNIST are not really the best data sets to try in FL (although they are commonly used). (ii) One of the reviewers had several critical comments about the theoretical results, please address those in the updated version. (iii) Please clarify in more detail how the theoretical and algorithmic contributions of there paper go beyond the recent work of (mirzasoleiman et el. 2020); (iv) iIt seems to me that the paper is missing some references on client selection in federated learning. Please revise the related work accordingly.